# Influence of Different Stabilization Systems and Multiple Ultraviolet A (UVA) Aging/Recycling Steps on Physicochemical, Mechanical, Colorimetric, and Thermal-Oxidative Properties of ABS

**DOI:** 10.3390/ma13010212

**Published:** 2020-01-04

**Authors:** Rudinei Fiorio, Sara Villanueva Díez, Alberto Sánchez, Dagmar R. D’hooge, Ludwig Cardon

**Affiliations:** 1Centre for Polymer and Material Technologies (CPMT), Department of Materials, Textiles and Chemical Engineering, Ghent University, Technologiepark 130, 9052 Zwijnaarde, Belgium; ludwig.cardon@ugent.be; 2Tecnalia, Sustainable Polymers, Building Technologies, Área Anardi 5, 20730 Azpeitia, Gipuzkoa, Spain; sara.villanueva@tecnalia.com (S.V.D.); alberto.sanchez@tecnalia.com (A.S.); 3Centre for Textiles Science and Engineering (CTSE), Department of Materials, Textiles and Chemical Engineering, Ghent University, Technologiepark 70a, 9052 Zwijnaarde, Belgium; dagmar.dhooge@ugent.be; 4Laboratory for Chemical Technology (LCT), Department of Materials, Textiles and Chemical Engineering, Ghent University, Technologiepark 125, 9052 Zwijnaarde, Belgium

**Keywords:** plastic waste, circularity, accelerated aging, recycling, stabilization

## Abstract

Commercially mass-polymerized acrylonitrile–butadiene–styrene (ABS) polymers, pristine or modified by stabilization systems, have been injection molded and repeatedly exposed to ultraviolet A (UVA) radiation, mechanical recycling, and extra injection molding steps to study the impact of such treatments on the physicochemical, mechanical, colorimetric, and thermal-oxidative characteristics. The work focus on mimicking the effect of solar radiation behind a window glass as relevant during the lifetime of ABS polymers incorporated in electrical and electronic equipment, and interior automotive parts by using UVA technique. The accelerated aging promotes degradation and embrittlement of the surface exposed to radiation and causes physical aging, deteriorating mechanical properties, with an expressive reduction of impact strength (unnotched: up to 900%; notched: up to 250%) and strain at break (>1000%), as well as an increase in the yellowing index (e.g., 600%). UV-exposition promotes a slight increase in the tensile modulus (e.g., 10%). The addition of antioxidants (AOs) leads to a limited stabilization during the first UVA aging, although the proper AO formulation increases the thermal-oxidative resistance during all the cycles. Mechanical recycling promotes an increase in strain at break and unnotched impact strength alongside a slight decrease in tensile modulus, due to disruption of the brittle surface and elimination of the physical aging.

## 1. Introduction

Acrylonitrile–butadiene–styrene (ABS) polymers are, among others, largely used for the manufacturing of automotive parts as well as electrical and electronic equipment [1]. As a result, ABS is one of the most common materials found in waste of end-of-life vehicles (ELV) and waste of electrical and electronic equipment (WEEE), with variations in molecular and microstructure due to different environmental experiences. For instance, many ABS products are (in)directly exposed to sunlight during their lifetime. Moreover, heat flux and mechanical stress are created in ABS-based materials during their processing, typically employing extrusion and injection molding techniques.

It is well known that the exposition to ultravilolet (UV) radiation, heat, and stress in the presence of oxygen can promote oxidative degradation [2,3,4,5], reducing the quality and affecting the circularity of polymers. In particular, polybutadiene (PB)-containing polymers such as ABS and high-impact polystyrene (HIPS) are susceptible to both thermal- and photo-oxidation, in which the PB phase is initially degraded, reducing the elastomeric properties of the rubbery phase and inducing discoloration and losses in impact strength and toughness [6,7,8,9]. The oxidation of PB also induces degradation of the polystyrene component in the styrene-acrylonitrile (SAN) phase [8,10]. Notably, several additives have been used to increase the stability of polymers during their active lifetime and processing, such as antioxidants (AOs) and light stabilizers [11,12]. These additives can trap emerging free radicals generated by heat, stress, UV radiation, etc. or unstable intermediate products (e.g., hydroperoxides), transforming them into stable products, and thus reducing degradation [12].

Numerous studies already covered the effects of aging and multiple recycling steps on ABS polymer properties. For example, Bokria and Schlick [3] investigated the photodegradation of ABS exposed to ultraviolet B (UVB) radiation and oxygen at 45 ${}^{\circ }$C and evaluated the aging effects by attenuated total reflectance Fourier transform infrared spectroscopy (ATR-FTIR). The authors concluded that degradation of ABS occurred due to the elimination of the double bonds of 1,4-butadiene, while the SAN matrix is unmodified. Major effects occurred in the exposed surface up to a depth of ca. 50 μm, where the butadiene structure was modified. In the unexposed surface, butadiene units were also thermally degraded. However, no degradation was observed in the center of the samples.

Pérez et al. [13] investigated the effects of reprocessing and combined recycling-accelerated aging based on UVB exposition for a wide portfolio of ABS polymer properties. The reprocessing cycles and the accelerated aging did not cause considerable modification with respect to thermal degradation, melt flow indexes, tensile moduli, and glass transition temperature ($T_{g}$) values. However, the reprocessing promoted a reduction in the impact strength, whereas the accelerated weathering combined with the reprocessing cycles caused a substantial reduction in the tensile strength.

Boldizar and Möller [14] studied the effect of multiple cycles of extrusion and aging in air regarding the properties of conventional ABS. These authors observed a pronounced reduction in the elongation at break due to the aging, followed by an increase of this property after extrusion. The reduction in the elongation at break was associated with the physical aging (annealing) of the SAN phase as well as to the thermal-oxidative degradation of the PB phase, mainly in the surface of the material. The temperature of oxidation presented a gradual decrease with an increase in aging and recycling steps, while the aged samples presented an increase in $T_{g}$ of the SAN phase with the increase in the number of cycles.

The effect of artificial and natural weathering on the photo-degradation of ABS was also studied by Santos et al. [15]. These authors observed that degradation occurred mainly on the surface, affecting optical, mechanical, and rheological properties. Both natural and artificial aging caused a substantial decrease in the strain at break during tensile tests as a consequence of the fragile and brittle degraded surface.

Despite the progress mentioned in the previous studies, the effects of different stabilization systems on the properties of ABS polymers submitted to multiple UV accelerated aging and recycling steps are not completely known. In view of the circularity of ABS polymers, many tests need to still be performed, specifically with ultraviolet A (UVA) light which simulates the solar radiation behind a window glass—a condition commonly found during the lifetime of electrical and electronic equipment, and interior automotive parts. Therefore, the aim of the present work is to investigate (i) ABS polymer degradation with different stabilization systems; and (ii) the relevance of multiple accelerated aging and recycling steps on the physicochemical, mechanical, colorimetric, and thermo-oxidative properties.

## 2. Materials and Methods

### 2.1. Materials

ABS Magnum 3453 Natural (Trinseo, Terneuzen, The Netherlands, obtained by mass polymerization), Irganox 1076 [octadecyl-3-(3,5-di-tert-butyl-4-hydroxyphenyl)propionate] (BASF, Ludwigshafen, Germany, phenolic primary antioxidant), Irganox 245 [ethylenebis(oxyethylene)bis-(3-(5-tert-butyl-4-hydroxy-m-tolyl)propionate)] (BASF, phenolic primary antioxidant), and Irgafos 168 [tris(2,4-ditert-butyl-phenyl)phosphite] (BASF, phosphite secondary antioxidant) were used as received unless stated otherwise.

### 2.2. Methods

#### 2.2.1. Extrusion-Based Manufacturing of Stabilization Masterbatches for First Injection Molding

Two masterbatches were prepared in a micro-compounder (Minilab II, Haake Thermo Fisher Scientific, Karlsruhe, Germany) at 220 ${}^{\circ }$C, 150 rpm, and at an extrusion rate of approximately 100 g·h${}^{-1}$. One masterbatch contained 5 m% of Irganox 1076 (Masterbatch 1) and the other masterbatch contained 5 m% of Irganox 1076, 5 m% of Irganox 245, and 5 m% of Irgafos 168 (Masterbatch 2). Before processing, the ABS (Magnum 3453 Natural) was dried at 80 ${}^{\circ }$C for 4h in a compressed-air dryer (CARD 40E, FarragTech, Wolfurt, Austria). Antioxidants (AOs) were used as received. After extrusion, the masterbatches were cooled by air at room temperature and granulated.

#### 2.2.2. Injection Molding

Three sample types, presenting different stabilization systems, were prepared by injection molding: pristine ABS (short notation: ABS 0), ABS containing 4 m% of Masterbatch 1 (short notation: ABS 1), and ABS containing 4 m% of Masterbatch 2 (short notation: ABS 2). The final mass contributions of the stabilizers used in each sample type are shown in Table 1. Before processing, the ABS and the masterbatches were dried at 80 ${}^{\circ }$C for 4 h in a compressed-air dryer. Tensile test specimens (according to ISO 527-1A standard [16]) and bars with dimensions of 100 mm × 10 mm × 4 mm were molded in an Engel E-Victory 28T (Schwertberg, Austria) injection molding machine using a temperature profile of 250–220 ${}^{\circ }$C (nozzle–hopper), a mold temperature of 60 ${}^{\circ }$C, an injection speed of 90 mm·s${}^{-1}$, an injection pressure of 820 bar, and a holding pressure of 410 bar.

#### 2.2.3. Accelerated Aging

The accelerated aging was conducted by exposing the samples to ultraviolet A (UVA) radiation (QUV/basic, Q-Panel Lab Products, Westlake, OH, USA) during 360 h at 50 ${}^{\circ }$C using a UVA-351 (type 1B) fluorescent UV lamp, which has a peak emission of λ = 353 nm. Such conditions are used for simulation of the UV portion of solar radiation behind window glass. This experiment was conducted according to the ISO 4982 standard [17].

#### 2.2.4. Mechanical Recycling

After the accelerated aging, the samples were shredded (RSP 15/30, Piovan, Santa Maria di Sala, Italy), dried (80 ${}^{\circ }$C, 4 h), and extruded (Plasti-Corder, Brabender, Duisburg, Germany). The extruder characteristics are screw diameter 19 mm, length/diameter ratio 25, barrel temperature profile 220–180 ${}^{\circ }$C (die–hopper), and screw speed 60 rpm. The extruded materials were cooled in a water bath at room temperature and granulated. After recycling, the samples were dried and injection molded as previously described, followed by another accelerated aging and mechanical recycling step. Figure 1 presents the research process flow chart, covering two repetitions as highlighted with the large grey arrow in the right part of this figure.

### 2.3. Characterization

#### 2.3.1. Chemical Characterization

Fourier transform infrared spectroscopy (FTIR; Bruker Tensor 27, Billerica, MA, USA) analyses were carried out in order to evaluate chemical modifications in the ABS polymer. Samples were studied in attenuated total reflectance (ATR) mode from 4000 to 600 cm${}^{-1}$. The values of absorbance were determined using the baseline method. An overview of relevant absorbances is provided in Table 2. All the absorbance results were normalized considering the maximum absorbance observed for the nitrile group (CN) at 2237 cm${}^{-1}$. This absorption band was chosen since the CN group is expected to be stable after aging and recycling steps [3,18,19,20].

Both the unexposed and exposed to UV radiation sides of the aged samples were analyzed. To evaluate the depth of the chemical modification caused by UV aging, the surfaces of both sides of the samples were microtomed (Leica RM2245 microtome, Wetzlar, Germany) and the chemical structure in different depths was investigated (depth = 0, 100, 200, and 300 μm). The carbonyl (CO, ca. 1730 cm${}^{-1}$), the 1,4-butadiene (1,4BD, 966 cm${}^{-1}$), and the nitrile (CN, 2237 cm${}^{-1}$) absorption peaks were evaluated using the absorption ratios CO/CN and 1,4BD/CN as defined by the Equations (1) and (2), respectively [20].
(1)$$\mathrm{CO}/\mathrm{CN}=\frac{\mathrm{Absorbance}\;\mathrm{at}\;1730\;{\mathrm{cm}}^{-1}}{\mathrm{Absorbance}\;\mathrm{at}\;2237\;{\mathrm{cm}}^{-1}}$$
(2)$$1,4\mathrm{BD}/\mathrm{CN}=\frac{\mathrm{Absorbance}\;\mathrm{at}\;966\;{\mathrm{cm}}^{-1}}{\mathrm{Absorbance}\;\mathrm{at}\;2237\;{\mathrm{cm}}^{-1}}$$

#### 2.3.2. Mechanical Properties

The impact strength (notched and unnotched) and the tensile properties of the injection molded and UV-aged samples were investigated according to the ISO 180 and ISO 527 standards, respectively [16,21].

#### 2.3.3. Morphological Characterization

The morphology of the aged injection molded samples was investigated by scanning electron microscopy (SEM) in a Phenom G1 microscope (FEI Company, Hillsboro, OR, USA), using an electron acceleration voltage of 5 kV. SEM micrographs were obtained from the surfaces exposed and unexposed to UVA. The SEM samples were obtained by cutting the aged parts at room temperature, using a cutting plier. The samples were dried in an oven at 80 ${}^{\circ }$C for 4 h and maintained in a desiccator until the analysis.

#### 2.3.4. Colorimetric Characterization

The color properties of the samples were investigated with an XPM spectrophotometer (Konica Minolta, Tokyo, Japan) using a D65 light source and 10${}^{\circ }$ viewing angle. All color measurements were obtained from the injection molded specimens. For the samples exposed to UV radiation, the color characteristics of both sides of the samples (exposed and unexposed sides) were analyzed. The yellowness index (YI) was used for evaluating the color modifications, as obtainable from the values of $L^{*}$ and $b^{*}$ (CIELAB color space) according to Equation (Equation 3) [22].
(3)$$YI=\frac{142.86.b^{*}}{L^{*}}$$

#### 2.3.5. Thermal-Oxidative Resistance

The thermal-oxidative resistance of all the samples was evaluated by determining the oxidation onset temperature (OOT) and the oxidation peak temperature (OPT). OOT and OPT indicate a relative degree of oxidative stability of a certain material [20]. A simultaneous thermal analyzer (STA 449 F3 Jupiter, Netzsch, Selb, Germany) was used, and the method was based on the ASTM E2009 standard [23]. For the aged samples, the thermal-oxidative resistance of both sides (exposed and unexposed to UV) were analyzed. Films of ~100 μm were obtained from the surfaces of the samples using a microtome. The experiments were conducted under an oxygen flow of 50 mL·min${}^{-1}$ and a sample mass of 4 ± 1.5 mg. A Pt-Rh pan was used, and the samples were studied from 50 to 500 ${}^{\circ }$C at a heating rate of 10 ${}^{\circ }$C·min${}^{-1}$.

## 3. Results and Discussion

### 3.1. Chemical Characterization

Figure 2 presents the FTIR spectra of the injection molded sample ABS 0 (no extra stabilizer system; entry 1 in Table 1), before aging (injection molded; black line), and after the 1st aging step with surfaces exposed (red line) and unexposed to UV (blue line). After UV aging, the chemical modification of the surfaces is noticeable. Figure 2 shows an increase in the absorption peaks at 1730 cm${}^{-1}$ (carbonyl, CO) and at 1600, 1580, 1070, and 1028 cm${}^{-1}$, indicating an oxidation of the sample, whereas the absorption peaks related to butadiene at 966 (1,4-butadiene) and 911 cm${}^{-1}$ (1,2-butadiene) decreased with the aging, representing degradation of the rubbery phase. The degradation of the side unexposed to UV can be related to thermal aging, since the accelerated aging cycle was conducted at a temperature of 50 ${}^{\circ }$C during 360 h. Similar results were observed in previous studies involving degradation of ABS [3,7,9,18,24,25,26,27]. However, some authors also identified an absorption band associated to hydroxyl groups (OH, ca. 3450 cm${}^{-1}$), which was not observed in the results of the present study. The OH absorption band was associated to alcohols and carboxylic acids [3,9,18,24,26], with alcohols resulting from the reaction of peroxy radicals with the polymer chains [18]. The observed discrepancy between the results of the present work and the previous ones may be related to the differences among the ABS materials studied, such as the PB content and the stabilizers used.

For completeness, note that all the samples showed similar results as in Figure 2, explaining why the focus was restricted to a limited number of experiments. For the additional spectra, the reader is referred to the Appendix A.

From Figure 2 (focus on ABS 0 and only the 1st aging step), the most evident modifications caused by UV exposition were observed in the carbonyl (CO) and 1,4-butadiene (1,4BD) absorption bands. Therefore, the CO/CN and 1,4BD/CN absorption ratios (Equations (1) and (2), respectively) were further investigated. These absorption ratios, obtained at different depths of the side exposed to UV radiation, are presented in Figure 3 covering results for all samples and various cycles. One can clearly notice that the exposition to UVA radiation increased the CO/CN and decreased the 1,4BD/CN absorption ratios at the surface (depth = 0 μm), whereas for larger depths (≥100 μm) these absorption ratios do not show a considerable modification compared to the values obtained for the unaged samples.

Moreover, the change in the stabilization system, as well as the increase in the number of aging cycles, do not promote an appreciable change in the absorption ratios. In this respect, the higher CO/CN absorption ratio observed for the sample ABS 1—3rd UV aging was considered as insignificant due to the high standard error found for this sample. Therefore, the degradation promoted by the UVA radiation was superficial (<100 μm in depth). This superficial degradation was also previously observed [3,14,15], and it is associated to the limited oxygen diffusion into the ABS and to a decrease of the absorbed light intensity through the polymer [8,9,18]. Changing the color characteristics due to aging could also modify the absorbance of the light and affect the degradation [18]. However, the modification of the color properties with the consecutive UV aging and recycling steps, as discussed below, does not alter the FTIR results observed in this work.

### 3.2. Mechanical Properties

#### 3.2.1. Impact Strength

Figure 4 shows the results for the notched (Figure 4a; lower starting values) and unnotched (Figure 4b; higher starting values) impact resistance, respectively, covering the three sample types described in Table 1 and several cycles of recycling and aging, as displayed in Figure 1. One can observe in Figure 4a that the addition of AOs does not change the notched impact strength. However, in Figure 4b a higher unnotched impact resistance is observed for the sample types containing AOs (ABS 1 and ABS 2 in Table 1) after the 1st UV exposition, showing a stabilization effect promoted by the presence of AOs.

The exposition to UV and the subsequent mechanical recycling substantially reduces both the notched (changes up to 250%) and unnotched (changes up to 900%) impact strength, indicating that degradation processes occurred during both accelerated aging and mechanical recycling [3,14,15,28,29]. A remarkable difference is noticed comparing the notched and unnotched impact strength, between the 2nd aging and the 2nd mechanical recycling treatments, as well as between the 2nd recycling and the 3rd aging, with the notched impact strength results similar for the 2nd aging and the 2nd recycling treatment. However, the unnotched impact strength is considerably higher after the 2nd mechanical recycling, in comparison with the results found for both 2nd and 3rd accelerated aged samples.

Similar behavior was observed by Boldizar and Möller [14] for the elongation at break during tensile tests, in which this elongation increased after the extrusion of accelerated aged ABS polymers. These authors associated the changes to both reversible physical aging and thermo-oxidative degradation. Moreover, Wysgoski [25] observed that thermal treatments by oven aging can promote embrittlement of ABS even without oxidation and correlated this behavior to conformational changes in the SAN phase due to annealing. Another possible explanation for the unnotched impact results is the formation of a degraded, brittle surface during the UV aging, which reduces the overall impact strength of the ABS-based material. This brittle surface is not present in the mechanically recycled samples since it is disrupted and homogeneously distributed in the sample during the shredding, extrusion and injection molding steps. Therefore, mechanical recycling removes the brittle surface, which can induce crack propagation [14,30], as well as eliminates the physical aging, thus increasing the unnotched impact strength.

#### 3.2.2. Tensile Properties

Figure 5 shows the tensile modulus (Figure 5a) and strain at break (Figure 5b) results according to the same ordering as in Figure 4. From Figure 5a, it is observed that the 1st aging causes an increase in the tensile modulus of the sample ABS 0 (unmodified sample type), indicating modification. With the addition of AOs the UV resistance of the ABS slightly increased during the 1st aging, and the tensile modulus does not change. Both 1st and 2nd mechanical recycling step induce tensile modulus values close to those found for the injection molded (unaged) samples. The 2nd and 3rd aging steps cause an increase in the tensile modulus for all samples. The accelerated aging promotes physical aging such as conformational changes in the SAN phase, as well as results in the formation of a brittle, degraded surface, which causes the increase in the tensile modulus values. As explained above, the recycling steps remove the effect of the physical aging, changing again the conformation of the SAN molecules [25], as well as eliminates the brittle surface, thus lowering the tensile modulus. Overall, the changes in the tensile modulus are between 5 and 10%.

Figure 5b shows the strain at break for the same samples as in Figure 5a. It is observed that the addition of AOs causes a reduction in the strain at break for the unaged (injection molded) samples, indicating that these additives can act as impurities, reducing the toughness of ABS. UV aging promotes a drastic reduction in the strain at break (factor > 10), which occurs before the yield point for all UV-aged samples. The reduction in the strain at break due to accelerated aging and reprocessing of ABS was previously observed in other studies [14,15]. Once again, the dramatic reduction in the elongation at break is related to the physical aging and embrittlement of the surface. This fragile surface can induce crack propagation during the tensile deformation [15], reducing the overall toughness. The addition of AOs does not promote any noticeable modification of this property.

Compared to the UV-exposed samples, mechanical recycling leads to an increase in the strain at break since the recycling generated a more homogeneous material. However, the 2nd recycling promotes a reduction in the strain at break, compared to the result found for the 1st recycling step. Degradation occurs during the 2nd aging as well. Moreover, one can expect an increase in the concentration of degraded polymer from the aged surface, distributed in the material after the 2nd recycling, which can induce crack propagation and reduce the toughness.

Table 3 shows the tensile strength of the sample types. It is observed that the 1st aging promotes a slight reduction of this property. However, the consecutive treatments do not promote a noticeable modification of the tensile strength. Rahimi et al. [29] also observed small changes (up to 3%) in the tensile strength of ABS after five reprocessing cycles by injection molding. Contrarily, Pérez et al. [13] observed a substantial decrease in the tensile strength with the increase in UVB exposition time and correlated the results to crosslinking of the PB phase. The type of ABS used in this work is different than the one used by Pérez et al., and this may be the reason for the opposite results. Therefore, the tensile strength property is not able to indicate the degradation occurred due to the multiple UV aging and reprocessing steps.

### 3.3. Morphological Characterization

Figure 6 shows the scanning electron microscopy (SEM) images for the exposed (Figure 6a) and unexposed (Figure 6b) sides of ABS 2 (entry 3 in Table 1) after the 1st aging step. One can observe that the exposed surface (Figure 6a) presents micro-cracks close to the edges. These micro-cracks are not observed on the unexposed side (Figure 6b). The same behavior is observed for all the other samples (see Appendix A). The micro-cracks are formed during the cut of the samples and are observed only on the surfaces exposed to UV, as these surfaces become brittle with the degradation, as highlighted above. Iannuzzi et al. [27] observed that the surface micro-hardness increases with the aging time of ABS due to degradation, which corroborates the embrittlement observed in the present study. Besides the micro-cracks observed, no other differences between the aged and unaged surfaces are observed.

### 3.4. Colorimetric Characterization

Figure 7a shows the yellowing indexes (YI values) for the studied sample types from Table 1. It is observed that both accelerated aging and mechanical recycling steps cause an increase in YI, indicating that these treatments cause degradation of the polymer. Photo-oxidation of ABS increases the UV and visible light absorption, increasing the yellowing of the material [18]. The addition of AOs does not promote any noticeable stabilization of the color characteristics. The 1st recycling step promotes a large increase in the YI, possibly related to the disruption of the oxidized surface during shredding, extrusion and injection molding, which increases the surface area of the degraded superficial layer and contributes to an increase in YI.

The extrusion and injection molding of ABS induces oxidation and modifies the color characteristics, as observed by Karahaliou and Tarantili [26]. After the 2nd aging, the YI values are similar to those found for the 1st recycling step, though the 2nd recycling promotes a further increase in YI. However, the 3rd accelerated aging causes a reduction in the YI values, compared to those found after the 2nd recycling. This reduction may be related to the formation of new chromophore groups with different color characteristics, which can cause fading [9]. Furthermore, UV aging can reduce the surface roughness of polymers [31] or induce blistering formation [9], which can also lead to color modifications.

As shown in Figure 7b, for the unexposed side, a slight increase in YI is observed after the 1st aging. This increase in YI values is likely related to oxidation promoted by the thermal aging (annealing at 50 ${}^{\circ }$C). This oxidation of the unexposed side is also observed in the FTIR results. Note that a sole focus on YI values is thus not recommended as each macroscopic property has a different sensitivity toward molecular and microstructural variations.

### 3.5. Thermal-Oxidative Resistance

Figure 8 shows the corresponding oxidation onset temperature (OOT— Figure 8a) and oxidation peak temperature (OPT—Figure 8b) data, including data for the sides exposed and unexposed to UV. The OOT values are obtained in the beginning of the oxidative process, while the OPT is associated to the maximum oxidation rate [20]. The higher the OOT and OPT values, the more stable the polymer is to thermal-oxidative degradation. The heat flux curves used for the determination of OOT and OPT are shown in the Appendix A. From Figure 8, one can see that the injection molding of ABS 0 promotes a slight decrease in both OOT and OPT compared to the results found for the unprocessed (virgin) ABS (first extra data point). This expected decrease in the thermal-oxidative resistance is related to the consumption of AOs added by the polymer manufacturer and degradation occurring during processing [12,20]. The addition of AOs increases the thermal-oxidative resistance of ABS, as previously observed [20]. ABS 1 presents slightly higher OOT and OPT values compared to those found for ABS 0, whereas ABS 2 showed a remarkably higher thermal-oxidative resistance, highlighting the relevance of the selection of the most appropriate mixture of stabilizers.

The effect of the sequential aging and recycling treatments on the OOT and OPT are also shown in Figure 8, with lines added for visual guidance. The smallest thermal-oxidative stability is observed for ABS 0, while the highest one is found for ABS 2. However, one can clearly observe that the accelerated aging promotes a reduction in the thermal-oxidative stability for both exposed and unexposed sides, while a subsequent mechanical recycling step increases the oxidative stability. Furthermore, it is observed that the stabilization systems used in the samples ABS 1 and ABS 2 are effective during all treatments, lasting up to the 3rd aging step.

It is also observed that after the 1st UV aging, the exposed sides of the samples present higher OOT and OPT values than those found for the unexposed sides. This higher thermal-oxidative stability of the exposed sides is likely related to the higher level of degradation detected for these surfaces; see the FTIR and YI results (Figure 2 and Figure 7, respectively). These highly oxidized surfaces may present a lower oxygen diffusion rate, which should induce a higher oxidative resistance, increasing both OOT and OPT. Moreover, the reduction in the thermal-oxidative resistance of the unexposed sides are probably associated to the consumption of AOs during the aging. It is expected that aged surfaces present a lower concentration of AOs compared to the core of the sample since the AOs near the surfaces are consumed during accelerated aging [12]. The mechanical recycling promotes an increase in the homogeneity of the samples, removing the degraded surfaces and homogeneously distributing the remaining AOs in the polymer, which increases both OOT and OPT.

Figure 8 also shows that the differences between the exposed and unexposed sides for both OOT and OPT values become smaller after the 2nd and 3rd aging, compared to the results found for the 1st aging treatment. These results may be associated with an increase of the thermal-oxidative resistance of the unexposed side after the 1st aging and 1st recycling steps due to the formation of more stable compounds after the treatments, or due the formation of products that can show a stabilizing effect, as suggested by Camacho and Karlsson [32].

Boldizar et al. [14] observed a gradual decrease in the temperature of oxidation with the increase in oven-aging and recycling steps associated with the consumption of stabilizers. The differences between the results found in the present study and the previous one may be related to the different aging conditions (UVA vs. oven), as well as different types of ABS and stabilization systems.

## 4. Conclusions

In the present study, the effect of stabilization systems and consecutive UVA aging followed by mechanical recycling and injection molding on the properties of commercial ABS is investigated.

The UVA aging promotes a severe degradation of the irradiated surface (<100 μm), as well as induced to physical aging (annealing). Both the embrittlement of the surfaces and the physical aging are responsible for the deterioration of the mechanical properties of the ABS. However, the core of the material, the major fraction of the polymer, does not present noteworthy degradation.

Accelerated aging and recycling steps substantially reduce the toughness of ABS. Particularly, the presence of a notch drastically reduces the impact strength, showing the importance of the design on the toughness of recycled plastic parts.

The mechanical recycling of the samples, conducted after the accelerated aging, leads to a considerable increase in the strain at break and unnotched impact strength, in addition to a slight decrease in the tensile modulus, due to disruption of the brittle surface and elimination of the thermal aging. Therefore, recycled ABS can be used for the manufacturing of new plastic parts in which high impact strength, toughness, and clear colors are not a stringent requirement.

The addition of the appropriate mixture of stabilizers increases the thermal-oxidative resistance of ABS for several aging and mechanical recycling steps, showing the endurance of the AOs. Therefore, as the correct stabilization system is added before the manufacture of ABS plastic parts, the thermal-oxidative resistance can remain adequate during service life and various recycling steps, contributing to the maintenance of the properties of the polymer and to the circular economy.

## Figures and Tables

**Figure 1 materials-13-00212-f001:**
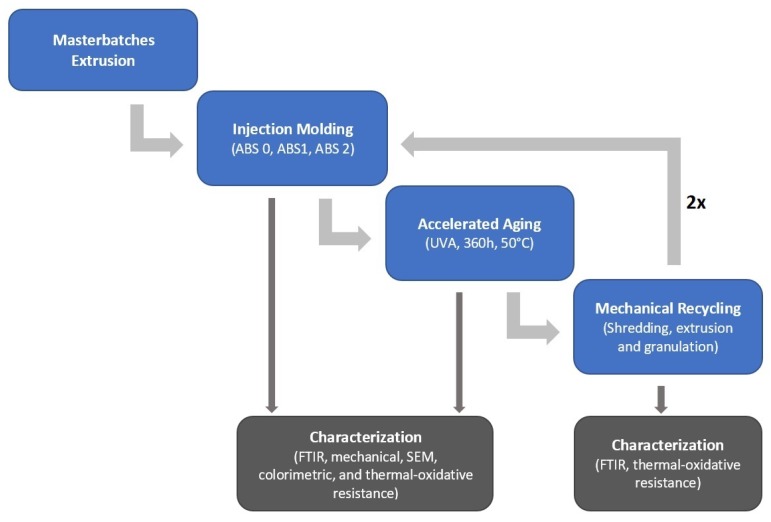
Research process flow chart to study the impact of various stabilization systems as combined with commercial ABS polymer (Table 1) after several accelerated aging and mechanical recycling steps.

**Figure 2 materials-13-00212-f002:**
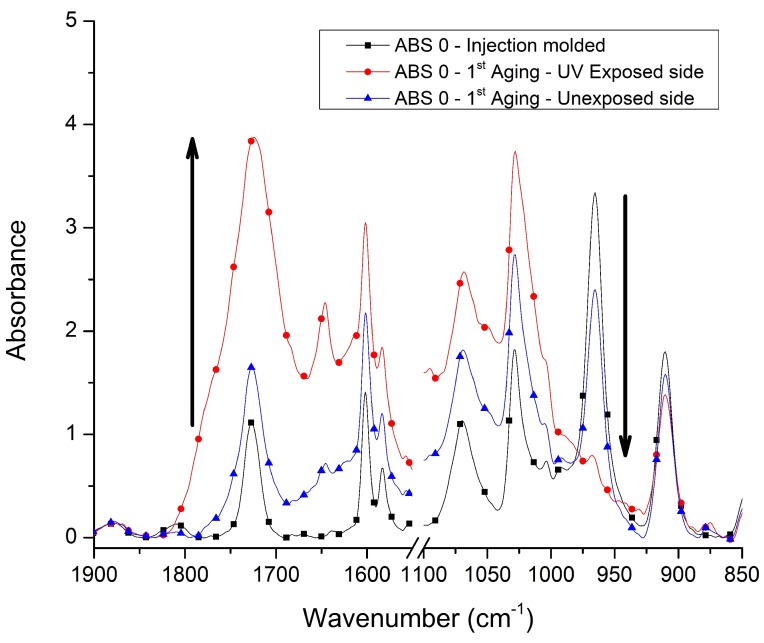
FTIR spectra of the sample ABS 0 (no extra stabilizer system; entry 1 in Table 1): injection molded (unaged) and after the 1st aging step (surfaces exposed and unexposed to UV). For other samples, see Appendix A.

**Figure 3 materials-13-00212-f003:**
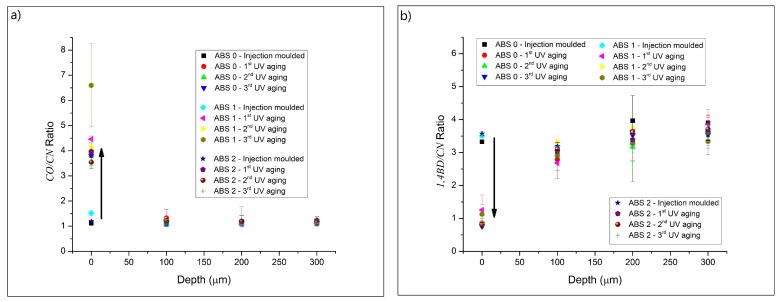
Absorption ratios for the side exposed to UV at different depths (Depth 0 = surface of the samples) for samples in Table 1 (all three entries) and up to two extra cycles; (**a**) CO/CN ratio, and (**b**) 1,4BD/CN ratio (Equations (1) and (2)). Substantial degradation is observed only at the surface.

**Figure 4 materials-13-00212-f004:**
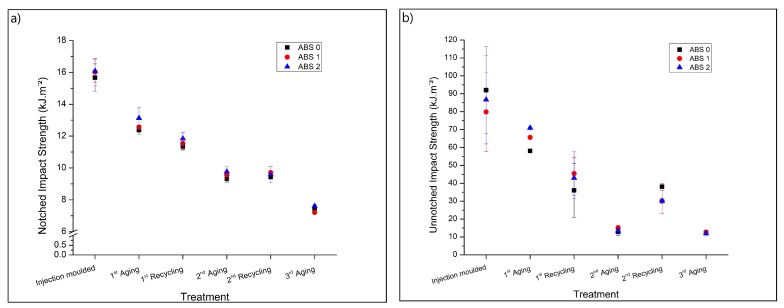
Impact strength of the sample types in Table 1. (**a**) Notched, and (**b**) Unnotched; testing procedure as in Figure 1.

**Figure 5 materials-13-00212-f005:**
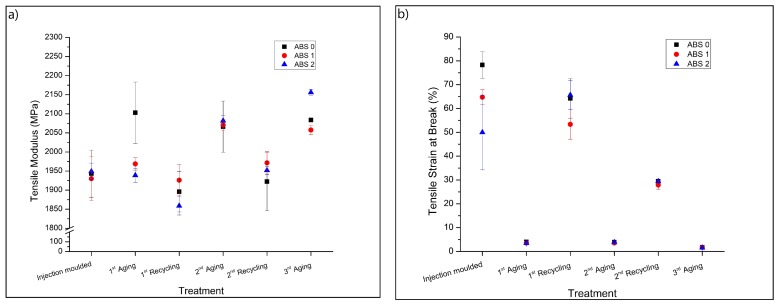
Tensile properties of the samples. (**a**) Tensile modulus, and (**b**) strain at break.

**Figure 6 materials-13-00212-f006:**
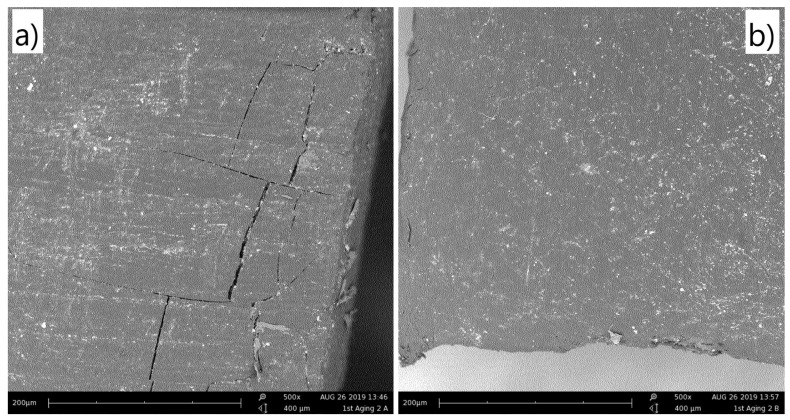
SEM images of the sample type ABS 2 (entry 3 in Table 1) after the 1st aging step. (**a**) UV exposed side and (**b**) unexposed side.

**Figure 7 materials-13-00212-f007:**
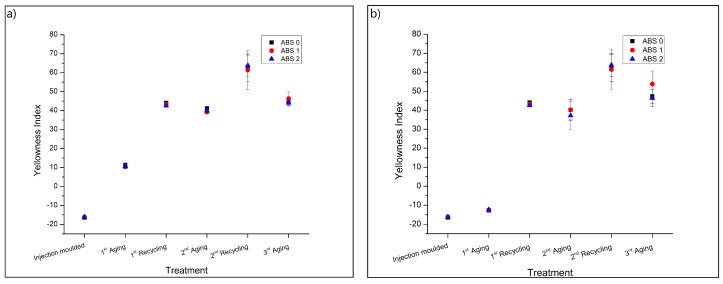
Yellowing index (YI) measurements. (**a**) UV-exposed side and (**b**) unexposed side for the sample types covered in Table 1, following the procedure in Figure 1.

**Figure 8 materials-13-00212-f008:**
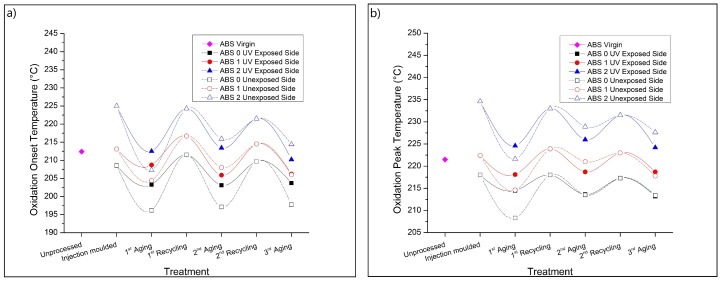
Thermal-oxidative resistance. (**a**) Oxidation onset temperatures and (**b**) oxidation peak temperatures for the samples covered in Table 1, following the procedure in Figure 1. Also added is the virgin ABS data point.

**Table 1 materials-13-00212-t001:** Final mass contributions of stabilizers in sample types for first injection molding step.

Sample Type	Masterbatch ${}^{a}$	Irganox 1076 (m%)	Irganox 245 (m%)	Irgafos 168 (m%)
ABS 0	-	-	-	-
ABS 1	1	0.2	-	-
ABS 2	2	0.2	0.2	0.2

${}^{a}$ Masterbatch 1 contained 5 m% of Irganox 1076, and Masterbatch 2 contained 5 m% of Irganox 1076, 5 m% of Irganox 245, and 5 m% of Irgafos 168.

**Table 2 materials-13-00212-t002:** Overview of relevant absorbances for the FTIR spectra.

Absorbance (cm${}^{-1}$)	Group
2237	Nitrile (CN)
1730	Carbonyl (CO)
966	1,4-Butadiene (1,4BD)
911	1,2-Butadiene (1,2BD)

**Table 3 materials-13-00212-t003:** Overview of tensile strength of the sample types as covered in Table 1.

Treatment	Tensile Strength (MPa)		
	ABS 0	ABS 1	ABS 2
Injection molded	32.2 ± 0.2	31.8 ± 0.1	31.9 ± 0.1
1st Aging	28.8 ± 2.3	27.4 ± 1.4	26.5 ± 1.2
1st Recycling	30.9 ± 0.1	30.6 ± 0.1	31.0 ± 0.1
2nd Aging	29.8 ± 0.1	28.7 ± 0.7	29.5 ± 0.3
2nd Recycling	30.8 ± 0.1	30.2 ± 0.3	30.4 ± 0.5
3rd Aging	32.0 ± 0.1	31.0 ± 0.9	30.8 ± 0.1

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
