# Peer review of "Influence of Different Stabilization Systems and Multiple Ultraviolet A (UVA) Aging/Recycling Steps on Physicochemical, Mechanical, Colorimetric, and Thermal-Oxidative Properties of ABS"

_materials, 2020, doi:10.3390/ma13010212_

Round 1

Reviewer 1 Report

The writing needs to be greatly improved. Some of the sentences, including the title, are extremely long and it is quite hard to follow these long sentences.  

The novelty or importance of the work is not clear; throughout the paper, it seems the authors are validating/comparing their results with some other group's work. It appears that all these results are published elsewhere and the authors seem to just "review" these reported results. 

Author Response

The authors are very grateful to the reviewers for their careful and meticulous reading of the manuscript. The reviews are detailed and helpful to finalize the manuscript. The authors would like to kindly acknowledge them. Here follow our comments to the major concerns and answers to specific points.

Reviewer 1 Comments

Point 1: The writing needs to be greatly improved. Some of the sentences, including the title, are extremely long and it is quite hard to follow these long sentences.

Response 1: The writing was checked and improved. The title was modified.

Point 2: The novelty or importance of the work is not clear; throughout the paper, it seems the authors are validating/comparing their results with some other group's work. It appears that all these results are published elsewhere and the authors seem to just "review" these reported results.

Response 2: The authors highlighted the novelty and importance of the work in the last paragraph of the Introduction section, which was modified:

“Despite the progress mentioned in the previous studies, the effects of different stabilization systems on the properties of ABS polymers submitted to multiple UV accelerated aging and recycling steps are not completely known. In view of the circularity of ABS polymers, many tests need to still be performed, specifically with ultraviolet A (UVA) radiation which simulates the solar radiation behind a window glass – a condition commonly found during the lifetime of electrical and electronic equipment, and interior automotive parts. Therefore, the aim of the present work is to investigate (i) ABS polymer degradation with different stabilization systems; and (ii) the relevance of multiple accelerated aging and recycling steps on the physicochemical, mechanical, colorimetric, and thermo-oxidative properties.”

Attached are the responses to all the reviewers, the revised manuscript and the supporting information files.

Sincerely,

The authors.

Reviewer 2 Report

The authors focus on investigating (i) ABS polymer degradation with different stabilization systems and employing repeated UVA aging; and (ii) the relevance of subsequent recycling steps on the physicochemical, mechanical, colorimetric, and thermo-oxidative properties. The results are acceptable, and the conclusion reflects the results reported. I think the manuscript is appropriate for publication in materials. However, the authors must consider the comments below.

Lots of grammatical errors occur in the main text. For example, Page 1, lines 5-7, the sentence “The focus is on UVA to mimic the effect of solar radiation behind a window glass as relevant during the lifetime of ABS polymers incorporated in electrical and electronic equipment, and interior automotive parts.” should be changed into “The work focus on mimicking the effect of solar radiation behind a window glass as relevant during the lifetime of ABS polymers incorporated in electrical and electronic equipment, and interior automotive parts by using UVA technique.”; Page 1, line 8, the word “modifying” should be “deteriorating”, because the word “modifying” means the good result.; Page 2, line 33, the word “as” should be deleted; Page 2, line 35, the sentence “…them into stable products, reducing degradation…” should be “…them into stable products, and thus reducing degradation…”; Page 3, line 73-75, the sentence “It should be  stressed  that  despite  the  progresses  mentioned  above  the  effects  of  multiple accelerated UV aging and mechanical recycling steps on the properties of ABS polymers focusing on various stabilization systems is still limited.” is not complete, please check it. Page 1, line 2, the authors should provide the full name of “UVA”. Page 7, about the Y-axis in Figure 2, the word “Absorbance (a.u.)” should be “Absorbance”. Page 9, lines 264-267, the authors claim that Pérez et al. observed a substantial decrease in the tensile strength with the increase in UVB exposition time and correlated the results to crosslinking of the PB phase. However, it is generally accepted that crosslinking effect leads to an increase in the tensile strength. Moreover, the authors do not provide the explanation why the tensile strength cannot change compared with that of injection molded samples after 3rd aging about Table 3. Page 11, line 300, the authors should define oxidation onset temperature (OOT). In addition, the authors should provide the thermal-oxidative degradation curves. More importantly, the theory values of OOT should be compared with its experiment values. Here related literatures should be referred and cited: Strengthening, toughing and thermally stable ultra-thin MXene nanosheets/polypropylene nanocomposites via nanoconfinement, Chemical Engineering Journal, 2019, 378: 122267; Bioinspired Design of Strong, Tough and Thermally Stable Polymeric Materials via Nanoconfinement, ACS Nano, 2018,12(9), 9266–9278. Kindly revise the language in the entire manuscript carefully.

Author Response

The authors are very grateful to the reviewers for their careful and meticulous reading of the manuscript. The reviews are detailed and helpful to finalize the manuscript. The authors would like to kindly acknowledge them. Here follow our comments to the major concerns and answers to specific points.

Reviewer 2 Comments

The authors focus on investigating (i) ABS polymer degradation with different stabilization systems and employing repeated UVA aging; and (ii) the relevance of subsequent recycling steps on the physicochemical, mechanical, colorimetric, and thermo-oxidative properties. The results are acceptable, and the conclusion reflects the results reported. I think the manuscript is appropriate for publication in materials. However, the authors must consider the comments below.

Lots of grammatical errors occur in the main text. For example,

Point 1: Page 1, lines 5-7, the sentence “The focus is on UVA to mimic the effect of solar radiation behind a window glass as relevant during the lifetime of ABS polymers incorporated in electrical and electronic equipment, and interior automotive parts.” should be changed into “The work focus on mimicking the effect of solar radiation behind a window glass as relevant during the lifetime of ABS polymers incorporated in electrical and electronic equipment, and interior automotive parts by using UVA technique.”;

Response 1: The sentence was modified according to the suggestion.

Point 2: Page 1, line 8, the word “modifying” should be “deteriorating”, because the word “modifying” means the good result.;

Response 2: The word was modified according to the suggestion.

Point 3: Page 2, line 33, the word “as” should be deleted;

Response 3: The word was deleted.

Point 4: Page 2, line 35, the sentence “…them into stable products, reducing degradation…” should be “…them into stable products, and thus reducing degradation…”;

Response 4: The sentence was modified according to the suggestion.

Point 5: Page 3, line 73-75, the sentence “It should be stressed that despite the progresses mentioned above the effects of multiple accelerated UV aging and mechanical recycling steps on the properties of ABS polymers focusing on various stabilization systems is still limited.” is not complete, please check it.

Response 5: The sentence was modified.

Point 6: Page 1, line 2, the authors should provide the full name of “UVA”.

Response 6: All the acronyms ‘UV’, ‘UVA’ and ‘UVB’ were written in full when they were used for the first time in the manuscript.

Point 7: Page 7, about the Y-axis in Figure 2, the word “Absorbance (a.u.)” should be “Absorbance”.

Response 7: The text was modified in Figure 2.

Point 8: Page 9, lines 264-267, the authors claim that Pérez et al. observed a substantial decrease in the tensile strength with the increase in UVB exposition time and correlated the results to crosslinking of the PB phase. However, it is generally accepted that crosslinking effect leads to an increase in the tensile strength.

Response 8: The authors agree with the reviewer, in which crosslinking generally leads to an increase in tensile strength. However, Pérez et al. (ref 13) state in their study: “…In consequence, crosslinking reactions can take place in the polybutadiene phase causing higher fragility which results in a lower tensile strength”. Moreover, Rahimi et al. (ref 15) observed an increase of only 3% in the tensile strength after 5 reprocessing of ABS by injection molding. The results obtained by Rahimi were added in the discussion.

Point 9: Moreover, the authors do not provide the explanation why the tensile strength cannot change compared with that of injection molded samples after 3rd aging about Table 3.

Response 9: Related to results presented in Table 3, the authors observed that the tensile strength property is not very sensitive to the degradation caused by the multiple aging and recycling steps and could not be clearly detected. An explanation was added in the text: “Therefore, the tensile strength property is not able to indicate the degradation occurred due to the multiple UV aging and reprocessing steps”.

Point 10: Page 11, line 300, the authors should define oxidation onset temperature (OOT).

Response 10: The term was defined in the section ‘Characterization - Thermal-Oxidative Resistance’: “OOT and OPT indicate a relative degree of oxidative stability of a certain material”.

Point 11: In addition, the authors should provide the thermal-oxidative degradation curves.

Response 11: The authors prepared another supporting information document showing all the thermal-oxidative resistance results (heat flux curves) to support the manuscript.

Point 12: More importantly, the theory values of OOT should be compared with its experiment values. Here related literatures should be referred and cited: Strengthening, toughing and thermally stable ultra-thin MXene nanosheets/polypropylene nanocomposites via nanoconfinement, Chemical Engineering Journal, 2019, 378: 122267; Bioinspired Design of Strong, Tough and Thermally Stable Polymeric Materials via Nanoconfinement, ACS Nano, 2018,12(9), 9266–9278.

Response 12: The authors appreciate the reviewer’s suggestions. However, the suggested literatures did not investigate the oxidation onset temperatures (OOT) or oxidation peak temperatures (OPT) of polymers. Moreover, the recommended literatures explore thermogravimetric analysis (TGA) data applying the rule of mixtures (ROM) to obtain theoretical values. The authors do not have the results of the OOT analyses for the pristine antioxidants or masterbatches used, necessary for the estimation of theoretical values by ROM. Hence, the authors are not able to conduct the determination of theoretical values of OOT and OPT.

Point 13: Kindly revise the language in the entire manuscript carefully.

Response 13: The language was revised.

Attached are the responses to all the reviewers, the revised manuscript and the supporting information files.

Sincerely,

The authors.

Reviewer 3 Report

This manuscript is a further investigation by authors on properties of  AO stabilized ABS polymers following their previous work on statistical analysis of similar effects published in Polymers (MDPI). While AO stabilization of ABS polymer is well studied, authors have introduced a novel aspect by investigating the effect of UV on AO stabilized ABS with different aging cycle and different levels of exposure. This is an interesting study and deserves publication. However, I will suggest few minor revisions-

# The title of the manuscript is very elaborated and must be modified to reflect the totality of the study in a more precise and brief manner.  

# Line 63: First sentence looks incomplete here.

# Overall, the introduction part has been slightly longer with too frequent referencing to related literature. I would recommend limiting this to the selected most relevant references.

# Why only two SEM images relating to only ABS 2 have been included? Including images for the other formulations and all aging steps as well as of injection molding would be appropriate.

# Conclusion section should be shorten to focus only on most important findings rather than discussing all of the experiments and should give a future direction for useful application of this information.

Author Response

The authors are very grateful to the reviewers for their careful and meticulous reading of the manuscript. The reviews are detailed and helpful to finalize the manuscript. The authors would like to kindly acknowledge them. Here follow our comments to the major concerns and answers to specific points.

Reviewer 3 Comments

This manuscript is a further investigation by authors on properties of  AO stabilized ABS polymers following their previous work on statistical analysis of similar effects published in Polymers (MDPI). While AO stabilization of ABS polymer is well studied, authors have introduced a novel aspect by investigating the effect of UV on AO stabilized ABS with different aging cycle and different levels of exposure. This is an interesting study and deserves publication. However, I will suggest few minor revisions-

Point 1: # The title of the manuscript is very elaborated and must be modified to reflect the totality of the study in a more precise and brief manner. 

Response 1: The Title was modified.

Point 2: # Line 63: First sentence looks incomplete here.

Response 2: The whole paragraph was removed, in order to reduce the length of the introduction.

Point 3: # Overall, the introduction part has been slightly longer with too frequent referencing to related literature. I would recommend limiting this to the selected most relevant references.

Response 3: The introduction was revised, and it was shortened.

Point 4: # Why only two SEM images relating to only ABS 2 have been included? Including images for the other formulations and all aging steps as well as of injection molding would be appropriate.

Response 4: Only the SEM images of Sample 2 were included in the manuscript because the edges of the sample are clear in these images. Moreover, all the other samples showed similar characteristics. The authors prepared another supplementary information document presenting the SEM images of all samples at a magnification of 500 times.

Point 5: # Conclusion section should be shorten to focus only on most important findings rather than discussing all of the experiments and should give a future direction for useful application of this information.

Response 5: The conclusions were reviewed and shortened.

Attached are the responses to all the reviewers, the revised manuscript and the supporting information files.

Sincerely,

The authors.

Reviewer 4 Report

The aim of work put forward by Fiorio et al., was investigating ABS polymer degradation with different stabilization systems and employing repeated UVA aging; and (ii) the relevance of subsequent recycling steps on the physicochemical, mechanical, colorimetric, and thermo-oxidative properties. Authors, focused on UVA to mimic the effect of solar radiation behind a window glass as relevant during the lifetime of ABS polymers incorporated in electrical and electronic equipment, and interior automotive parts.

The paper reports on an interesting issue in which important contributions are being performed recently. In general the paper is of above average value with respect to broadening knowledge and extending methods of composites characteristisc subjected to aging or recycling.

Noteworthy is the varied research methodology and the number of research experiments performed. The way the results are presented is clear and legible. Figures and tables understandable. The description of the results forms a unified whole creating a logical cause and effect sequence. Stylistics of language at a satisfactory level.

I recommend the paper to be publish in the Materials journal in its current form.

Author Response

The authors are very grateful to the reviewers for their careful and meticulous reading of the manuscript. The reviews are detailed and helpful to finalize the manuscript. The authors would like to kindly acknowledge them. Here follow our comments to the major concerns and answers to specific points.

Reviewer 4 Comments

The aim of work put forward by Fiorio et al., was investigating ABS polymer degradation with different stabilization systems and employing repeated UVA aging; and (ii) the relevance of subsequent recycling steps on the physicochemical, mechanical, colorimetric, and thermo-oxidative properties. Authors, focused on UVA to mimic the effect of solar radiation behind a window glass as relevant during the lifetime of ABS polymers incorporated in electrical and electronic equipment, and interior automotive parts.

The paper reports on an interesting issue in which important contributions are being performed recently. In general the paper is of above average value with respect to broadening knowledge and extending methods of composites characteristisc subjected to aging or recycling.

Noteworthy is the varied research methodology and the number of research experiments performed. The way the results are presented is clear and legible. Figures and tables understandable. The description of the results forms a unified whole creating a logical cause and effect sequence. Stylistics of language at a satisfactory level.

I recommend the paper to be publish in the Materials journal in its current form.

Response: The authors appreciate the reviewer’s recommendation.

Attached are the responses to all the reviewers, the revised manuscript and the supporting information files.

Sincerely,

The authors.

Round 2

Reviewer 1 Report

The authors have diligently addressed all the reviewers' comments.

Reviewer 2 Report

Due to authors carefully address the reviewers’ comments, I agree to recommend the acceptance of this revised manuscript.